# Venetoclax: A Game Changer in the Treatment of Younger AML Patients?

**DOI:** 10.3390/cancers16010073

**Published:** 2023-12-22

**Authors:** Matteo Molica, Salvatore Perrone, Vincenzo Federico, Caterina Alati, Stefano Molica, Marco Rossi

**Affiliations:** 1Department of Hematology-Oncology, Azienda Universitaria Ospedaliera Renato Dulbecco, 88100 Catanzaro, Italy; molica@bce.uniroma1.it (M.M.); mrossi@unicz.it (M.R.); 2Department of Hematology, Polo Universitario Pontino, S.M. Goretti Hospital, 04100 Latina, Italy; sperrone@hotmail.it; 3Hematology and Transplant Unit, Vito Fazzi Hospital, 73100 Lecce, Italy; federico.ematolecce@gmail.com; 4Hematology Unit, Department of Hemato-Oncology and Radiotherapy Grande Ospedale Metropolitano “Bianchi-Melacrino-Morelli”, 89124 Reggio Calabria, Italy; caterina.alati@gmail.com; 5Queens Centre for Oncology and Haematology, Castle Hill Hospital, Hull University NHS Trust, Hull HU16 5JQ, UK; 6Department of Experimental and Clinical Medicine, Magna Graecia University, 88100 Catanzaro, Italy

**Keywords:** acute myeloid leukemia, venetoclax plus intensive chemotherapy, high-risk patients

## Abstract

**Simple Summary:**

In recent studies, the combination of venetoclax with intensive chemotherapy has demonstrated encouraging results in the treatment of fit patients with de novo and relapsed/refractory acute myeloid leukemia (AML). This review focuses on ongoing clinical trials that investigated the addition of venetoclax to frontline regimens in younger AML patients and specific molecularly defined subgroups. Results of these trials indicate the potential benefits of integrating venetoclax into intensive chemotherapy strategies across European Leukemia Net (ELN) risk subcategories, thus offering valuable guidance for developing future clinical trials.

**Abstract:**

The combination approach based on venetoclax (VEN) with azacytidine (AZA) has significantly improved outcomes for elderly patients with acute myeloid leukemia (AML). This innovative approach has led to higher rates of overall response, measurable residual disease (MRD)-negative remissions, and overall survival compared with AZA monotherapy. As a result, this combination has emerged as the gold-standard treatment for elderly or unfit patients with AML who are not eligible for intensive therapy. In younger, fit patients with AML, intensive induction and consolidation chemotherapy is commonly used as a first-line approach; however, relapse continues to be the main reason for treatment failure in approximately 30–40% of patients. Efforts to improve MRD-negative response rates and to facilitate the transition to allogeneic hematopoietic stem cell transplantation, particularly in high-risk AML, have inspired trials exploring the combination of intensive chemotherapy with targeted agents. VEN, a first-in-class anti-BCL2 agent, combined with intensive chemotherapy regimens has shown deep MRD-negative remissions, producing prolonged event-free survival and enhancing the transition to allogeneic transplant in first-complete-remission patients. These benefits support the incremental advantages of adding VEN to intensive chemotherapy approaches across ELN risk subcategories, and provides a robust benchmark to design future trials. In this review, we will discuss current studies assessing the efficacy of frontline regimens integrating VEN into intensive chemotherapy in younger patients with AML and specific molecularly defined subgroups.

## 1. Introduction

Historically, anthracyclines and cytarabine-based regimens were the mainstay of frontline acute myeloid leukemia (AML) treatment, yielding complete remission (CR) rates of about 55% [1]. CR rates were raised to 65–78% by anthracycline selection and dose expansion [2,3,4], although mostly younger patients (i.e., under 50 years old) with favorable or intermediate-risk cytogenetics primarily benefited from these approaches [2]. Regretfully, relapses are still frequent and often occur after a median time of 24 months from diagnosis [1,2,3,4]. High-dose cytarabine consolidation or longer consolidation cycles have contributed to increasing the CR rates to about 75–85% [3,5,6]. Long-term event-free survival (EFS), however, continued to hover around 45% [4,5,6].

Measurable residual disease (MRD), assessed by either polymerase chain reaction (PCR) or multiparameter flow cytometry (MFC), has become a key biomarker for evaluating the effectiveness of AML treatment. Notably, MRD-negative remissions are correlated with a lower incidence of relapse and improved relapse-free survival (RFS) and overall survival (OS) in patients treated with less-intensive chemotherapy regimens (IC) [7,8,9,10], including consolidative HSCT. From a clinical standpoint, multi-drug IC regimens that induce MRD-negative remission in approximately 50–60% of patients [6,11], eventually leading to HSCT, are crucial for patients with an intermediate or unfavorable risk of de novo AML, secondary or therapy-related AML [12,13] and relapsed/refractory (R/R) AML [14,15].

In the phase 3 VIALE-A study [10], AML patients 75 years of age or older who were ineligible for intensive regimens due to coexisting comorbidities showed improved MRD-negative CR rates with venetoclax (VEN) combined with azacytidine (AZA) in comparison with placebo plus AZA (23.4% vs. 7.6%). However, it is still up for debate whether VEN plus intensive chemotherapy can ameliorate the response rate in younger and fitter patients, thereby producing a higher proportion of patients who undergo HSCT. In this review, we will examine ongoing trials that involve the addition of VEN to intensive chemotherapy frontline regimens, with a focus on evaluating its efficacy in younger patients with AML and among distinct molecularly defined subgroups. The literature search, conducted on PubMed up until 30 November 2023, aimed to recognize original full-text papers and research letters published in English. The search strategy utilized both Medical Subject Heading terms and free-text words to enhance its sensitivity. A complementary manual search was also conducted to screen the proceedings of the annual meetings of the American Society of Hematology, American Society of Clinical Oncology, and European Hematology Association from the last 5 years. Abstract evaluation and data extraction were independently performed by two reviewers (M.M. and S.P.), ensuring rigorous analysis.

## 2. Venetoclax Plus Intensive Chemotherapy in De Novo AML

VEN, a well-known BH3 inhibitor that selectively targets BCL2, plays a pivotal role in restoring the dysregulated pathway of apoptosis in various hematological cancers. BCL-2 expression has been shown to be significantly upregulated in newly diagnosed and relapsed AML patients (with a range of 34–87%). In vitro studies showed that VEN blocks BCL-2 activity, thus reducing the apoptotic threshold of AML cells and finally leading to an improved response to chemotherapy [16]. Based on these premises, a combinatorial treatment of VEN with chemotherapy represents a reasonable approach. Recently, many studies have assessed the combination of VEN with several IC regimens, including cytarabine and daunorubicin (DA), purine-analogue-based regimens (FLAG, FLAI, CLIA), and CPX351, both in adult and pediatric patients (Figure 1 and Table 1).

### 2.1. Venetoclax in Combination with Daunorubicin and Cytarabine

In a phase 2 study conducted at three public centers in China, 36 patients aged 18–60 years with untreated AML received induction treatment based on the combination of daunorubicin (60 mg/m^2^ on days 1–3), cytarabine (100 mg/m^2^ on days 1–7), and VEN (100 mg on day 4, 200 mg on day 5, and 400 mg on days 6–11; DAV regimen) [17]. Composite CR (CR plus CR with incomplete blood cell count recovery (CRi)) assessed after one cycle of induction treatment was the primary endpoint of the study. Following one cycle of DAV, 30 patients (91%) achieved a composite CR, with 29 (97%) showing undetectable MRD (<0.1%). Grade 3–4 toxicities comprised febrile neutropenia in 18 (55%), pneumonia in 7 (21%), and sepsis in 4 (12%) cases. No patient died from treatment-related complications. After a median follow-up of 11 months, the estimated 1-year OS was 97% and the 1-year EFS was 72%.

Another Chinese phase 2 trial investigated the combination of VEN and daunorubicin and cytarabine (DA 2  +  6) in 42 younger patients with de novo AML [18]. The primary endpoint of the study was the overall response rate (ORR), which included CR, CRi, and partial response (PR). The frontline chemotherapy included oral VEN (400 mg; days 1–7), daunorubicin (60 mg/m^2^; days 2–3), and cytarabine (200 mg/m^2^; days 2–7). The ORR and the composite CR rate (CR  +  CRi) following one cycle of induction were 92.9% and 90.5%, respectively; furthermore, 87.9% of the CR patients achieved an undetectable MRD. Grade 3–4 adverse events comprised neutropenia (100%), thrombocytopenia (100%), and febrile neutropenia (90.5%), and one induction death was observed. The median times of neutrophil and platelet recovery were 13 and 12 days, respectively. The estimated 1-year OS, EFS, and disease-free survival (DFS) rates were 83.1%, 82.7%, and 92.0%, respectively. Stratifying patients by the ELN2022 prognostic risk classification, the estimated 12-month OS, EFS, and DFS rates were 70.7%, 70.3%, and 79.9%, respectively, for adverse-risk patients.

The CAVEAT phase 1b trial enrolled patients aged ≥65 years diagnosed with de novo or secondary/therapy-related AML (s/t-AML) eligible for IC [19]. Five different VEN dose escalation groups (50–100–200–400–600 mg) were investigated. In two additional cohorts, the safety of reduced doses of VEN (50 or 100 mg) in combination with anti-fungal prophylaxis was also explored. The induction phase included 14 days of VEN in combination with “5 + 2” chemotherapy (cytarabine 100 mg/m^2^ on days 1–5 and idarubicin 12 mg/m^2^ on days 2–3) and consolidation with 14 days of VEN in combination with cytarabine 100 mg/m^2^ (days 1–2) and idarubicin 12 mg/m^2^ (day 1), followed by up to seven cycles of VEN monotherapy maintenance. The CAVEAT study includes one induction cycle followed by less-intensive post-remission approaches. To date, 69 patients have been enrolled, with a median age of 71 years. The CR/CRi rate was 73%, with 90% among de novo AML and 51% among s/t-AML patients. Interestingly, an adverse karyotype was found in 69% of responding patients. The median OS was 15.4 months in the overall patient population and 31.3 months in de novo AML cases. A subsequent analysis evaluated treatment-free remission (TFR), which was reported in 20 out of 51 (39%) patients, with the median TFR time being 22.1 months [19].

A real-life, single-center, retrospective experience including the “2 + 5” regimen in combination with VEN in patients ≥ 60 years was also reported by Wang et al. [20]. The 12 patients enrolled presented a median age of 64 years; among them, 33.3% (4/12) were classified as high-risk according to the ELN 2022 classification. The CR rate was 91.7%, with all patients classified as poor-risk achieving CR. Negative MRD was reached in all patients (100%) in CR. The most frequent ≥ 3 adverse events during induction were febrile neutropenia (41.7%) and pneumonia (33.3%). The estimated 1-year EFS and OS rates were 75% and 100%, respectively.

Based on a prior study showing that a 4-day treatment with cyclophosphamide and a 3-day treatment with cytarabine yielded a similar CR rate (80%) in comparison with the traditional 7-day regimen [21], Wang et al. designed a trial including VEN at doses of 200 mg per day from day 1 to day 7, cytarabine (1 g/m^2^) given every 12 h from day 1 to day 3, and cyclophosphamide given at 20 mg/kg/day from day 1 to day 3 (VCA regimen) [22]. Twenty-five patients (median age of 47.4 years) were included in this study. The patient class of risk, according to the ELN 2017, was as follows: 44% low-risk, 32% intermediate-risk, and 24% high-risk. Overall, 23 patients (92%) reached CR (including 1 CRi) after one cycle of VCA. Of note, all 23 CR patients (92%) experienced MRD negativity. Compared with the previous chemotherapy regimen (cytarabine plus cyclophosphamide), patients treated with VCA obtained a higher CR/MRD rate (92% vs. 45%; *p* < 0.01).

### 2.2. Venetoclax Plus Purine-Analogue-Based Regimens

The combination of VEN and FLAG-IDA (consisting of the combination of fludarabine, cytarabine, idarubicin, and granulocyte colony-stimulating factor) in induction and consolidation is currently under investigation in a phase 1b/2 trial enrolling de novo and R/R AML [23]. The induction with FLAG-IDA includes a 28-day cycle of fludarabine (30 mg/m^2^) associated with cytarabine (1.5–2 g/m^2^) on days 2–6, idarubicin (8 mg/m^2^ on days 4–6), and filgrastim (5 µg/kg on days 1–7). Consolidation consists of a shorter course of fludarabine and cytarabine (on days 2–4) and filgrastim (on days 1–5); idarubicin is permitted (on days 3–4) in up to two consolidation courses. At the recommended phase II dose, VEN is administered on days 1 to 14 in the induction phase (100 mg on day 1, 200 mg on day 2, 400 mg on day 3, and onwards; days 1–14) and on days 1–7 in consolidation. Following the completion of induction or consolidation, maintenance with continuous daily VEN is permitted in cycles of 1 to 28 days of each 28-day cycle for up to 1 year in patients not proceeding to HSCT.

To date, the phase 2 trial has enrolled 45 untreated patients with a median age of 44 years; among them, 73% had de novo AML, 16% sAML, and 11% therapy-related AML (t-AML). The ORR was 98%, and the median time to achieving the best response was 28 days [24]. No significant difference in MRD-negative achievement was documented across ELN risk groups (favorable, 100%; intermediate, 88%; and adverse, 94%); moreover, MRD-negative rates slightly differed between de novo and s/t-AML (95% vs. 90%). Overall, 60% of patients failed to proceed to HSCT after a median number of 3.4 cycles; the majority of them after 2 cycles of FLAG-IDA plus VEN. Among them, 58% were classified as adverse-risk at baseline. Overall, this purine analogue-based combination appeared safe, with no early death registered at days 30 and 60. Notably, a landmark analysis including patients in CR with a minimum of 3.4 months of study follow-up observed no significant difference in the median EFS (*p* = 0 .8) and median OS (*p* = 1.0) between patients proceeding to HSCT and those who did not. CR patients with a persistence of next-generation sequencing (NGS)-detected *TP53* mutations experienced an expansion of the founding clone at the time of relapse.

The GIMEMA (Gruppo Italiano Malattie Ematologiche dell’Adulto) group investigated in a phase 1/2 trial (AML1718 trial) the safety and efficacy of two different doses of VEN (400 and 600 mg) combined with the FLAI regimen (fludarabine 30 mg/m^2^ from day 1 to day 5, cytarabine 2000 mg/m^2^ from day 1 to day 5, and idarubicin 8 mg/m^2^ on days 1, 3, and 5) as a frontline approach in 57 untreated AML patients classified as ELN-2017 intermediate- or high-risk adult AML patients (median age 54 years; range 18–65) [25]. The dose of VEN was adjusted according to the concomitant posaconazole (CYP3A4 inhibitor) dose administered and was stopped until recovery for patients in response at day 21 and during consolidation; post-HSCT maintenance was not permitted. In this study, the CR rate was 84%, and notably, 74% of patients reached a negative MRD status after induction therapy. After a median follow-up of 10.5 months, 28 patients (80% in MRD-negative CR) were given allografted HSCT. The 1-year OS probability was 76%, while DFS was not reached. The 30-day and 60-day mortality was 1.8% and 5.3%, respectively. The patients treated with 400 mg and 600 mg of VEN showed no significant differences in terms of safety, CR, or survival. The potential benefit of the VEN-IC combination was assessed by comparing patient-level data from the GIMEMA AML1718 (V-FLAI group; *n* = 57) [25] to the “3 + 7”-like induction treatment (*n* = 445) [26] and to FLAI real-life experience (*n* = 155) [27]. According to a propensity-score-matching analysis [28], it was found that the CR rate in the V-FLAI group was superior (84%) to that of the FLAI and the “3  +  7” groups (80% and 63%, respectively).

Notably, the V-FLAI regimen was associated with a greater probability of reaching MRD-negativity compared with both the FLAI and “3  +  7” regimens (*p* = 0.005 vs. FLAI; *p* = 0.001 vs. “3  +  7”). The DFS and OS estimates of the V-FLAI cohort were superior compared with the other two regimens. However, a longer AML1718 follow-up is needed to confirm the superiority of the V-FLAI regimen.

A phase 2 clinical trial explored the combination of VEN with the CLIA regimen (cladribine 5 mg/m^2^ on days 1–5, cytarabine 1.5 g/m^2^ on days 1–5, and idarubicin 10 mg/m^2^ on days 1–3; VEN at a dose of 400 mg on days 2–8; consolidation consisted of 3 days of CLIA) in patients aged ≤65 years with untreated AML or high-risk MDS [29]. To date, the study has enrolled 67 patients, with 40% classified as high-risk according to the ELN2017 classification. The composite CR rate was 96%, with MRD-negativity by MFC observed in 77% of the patients. Interestingly, high-risk ELN patients achieved a composite CR of 96%, while the rate of response in patients with *TP53* mutations (*n* = 3) was 67%. The median number of administered cycles was 2, with 70% of patients proceeding to HSCT. At 24 and 12 months, the rate of patients in a leukemia-free state was 71% and 86%, respectively. A landmark analysis revealed that patients who underwent HSCT during their first remission experienced a longer OS (*p* = 0.036). There was only one reported early death, resulting in an 8-week mortality rate of 3%.

Recently, a post hoc propensity-score-matched analysis including different clinical trials (NCT03214562, NCT02115295, and NCT01289457) compared the activity of VEN plus IC vs. IC alone [30]. The analysis comprised 279 patients (median age 49 years): 85 in the VEN plus IC cohort (40 (47%) of 85 with FLAG-IDA plus VEN and 45 (53%) with CLIA plus VEN) and 194 in the IC cohort. Sixty-four (86%) of 74 patients in the VEN plus IC cohort achieved an MRD-negative composite CR compared with 86 (61%) of 140 patients from the IC cohort (*p* = 0.0028). Among the ELN adverse-risk group, patients treated with VEN plus IC showed a significant higher rate of negative MRD than patients treated with IC alone (87% vs. 48%; *p* = 0.0059). Overall, the rate of early mortality deaths was not significantly different between the two cohorts. Patients in the VEN plus IC arm carrying an adverse ELN risk proceeded more frequently to HSCT (82% vs. 46%; *p* < 0.0001). Finally, VEN plus IC was associated with better EFS (*p* = 0.030), while OS rates did not significantly differ between the two groups (*p* = 0.13).

### 2.3. CPX-351 Plus Venetoclax

An ongoing phase 2 trial is investigating the safety and efficacy of VEN in combination with CPX-351 (daunorubicin 44 mg/m^2^ and cytarabine 100 mg/m^2^ on days 1, 3, 5 of induction, and daunorubicin 29 mg/m^2^ + cytarabine 65 mg/m^2^ on days 1 and 3 during consolidation; the starting dose of VEN consists of 300 mg on days 2–21) in patients with newly diagnosed and R/R AML who are eligible for IC [31]. Among the five treatment-naïve AML patients, four (80%) obtained CR/CRi with an MRD-negativity of 75%. All four patients in CR or Cri proceeded to HSCT. The median OS among frontline AML patients was not reached, while the 1-year estimated OS was 75%.

A phase 1b study is assessing the safety and efficacy of lower-intensity CPX-351 and VEN in untreated AML patients who are ineligible for IC [32]. The trial included 31 patients: 4 patients in the dose-expansion phase (DEP) at dose level 1 (CPX-351 20 units/m^2^ on days 1 and 3 + VEN 400 mg on days 2 to 21 of each cycle), 7 patients in the DEP at dose level 2 (CPX-351 40 units/m^2^ + VEN 400 mg), and a total of 20 patients in the DEP and expansion phase at dose level 1b (CPX-351 30 units/m^2^ + VEN 400 mg). The median age of the patients was 74 years, and 58% presented poor-risk disease (23% with *TP53* mutations). No death was observed within 30 days from the therapy initiation; however, the mortality at 60 days was 13%. The CR/Cri rate was 57% (75% with MRD-negative disease), with all patients achieving remission after the first cycle.

**Table 1 cancers-16-00073-t001:** Trials including a venetoclax and chemotherapy combination as a frontline approach.

Trial/Reference	Design	Primary Endpoint	Number of Patients	Patients Enrolled	Response	Outcomes	Early Mortality
daunorubicin + cytarabine + venetoclax (DAV)/[17]	phase II	composite complete remission rate	36	patients aged 18–60 years	CRc ^6^ rate: 91%	estimated 1-year OS ^11^: 97% estimated 1-year EFS ^12^: 72%	30-day mortality: 0%
daunorubicin + cytarabine + venetoclax (DAV 2 + 6)/[18]	phase II	overall response rate	42	patients aged 16–60 years	ORR ^7^: 92.9%; 87.9% of the CR ^8^ patients with undetectable MRD ^9^	estimated 12-month OS ^11^: 83.1% estimated 12-month EFS ^12^: 82.7% estimated 12-month DFS ^13^: 92%	30-day mortality: 2.4%
daunorubicin + cytarabine + venetoclax (5 + 2 + VEN)/[19]	phase Ib	optimal dose schedule of venetoclax with 5 + 2	69	patients aged ≥65 years with de novo or s-AML ^1^ or t-AML ^2^	overall response (CR ^8^/Cri ^10^) rate: 73%	median OS ^11^: 15.4 months	30-day mortality: 6%
daunorubicin + cytarabine + venetoclax (5 + 2 + VEN)/[20]	retrospective clinical trial	composite complete remission	12	patients aged ≥ 60 years	CR ^8^ rate: 91.7% All patients with poor-risk achieved CR ^8^	estimated 1-year EFS ^12^: 75%. Estimated 1-year OS ^11^ rate: 100%	30-day mortality: 0%
cyclophosphamide + cytarabine + venetoclax (VCA)/[22]	pilot study	complete remission rate	25	adult AML ^3^	CR ^8^/Cri ^10^: 92%; all these patients had undetectable MRD ^9^	Estimated 12-month OS ^11^: 79.3%.	/
fludarabine + cytarabine + idarubicin + filgastrim + venetoclax (FLAG-IDA + VEN)/[24]	phase Ib/II	overall response rate	45	patients aged ≥18 (including de novo, sAML ^1^, tAML ^2^, tsAML ^4^, or high-risk MDS ^5^)	ORR ^7^: 98%; among CR ^8^ patients, 93% MRD ^9^ negative	estimated 24-month EFS ^12^: 64% estimated 24-month OS ^11^: 76%,	30-day mortality: 0% 60-day mortality: 0%
fludarabine + cytarabine + idarubicin + venetoclax (V-FLAI)/[28]	phase I/II trial	complete remission rate	57	European LeukemiaNet intermediate- or high-risk adult AML ^3^ (median age 54 years; 18–65)	CR ^8^ rate: 84%; MRD ^9^ negative: 74%	probability of 12-month OS ^11^: 76%	30-day mortality: 1.8% 60-day mortality 5.3%
cladribine + cytarabine + idarubin + venetoclax (CLIA + VEN)/[29]	phase II	complete response rate	67	patients aged ≤65 years with newly diagnosed AML ^3^ or high-risk MDS ^5^	CRc ^7^ rate: 96%; among CR ^8^ patients, 90% MRD ^9^ negative	estimated 12-month OS ^11^: 86.5% estimated 24-month OS ^11^: 86.5% estimated 12-month EFS ^12^: 71.8% estimated 24-month EFS ^12^: 69.7%	30-day mortality: 2% 60-day mortality 3%
CPX-351 + venetoclax (CPX-351 + VEN)/[31]	phase Ib/II	the safe dose and schedule	5	patients aged ≥ 18 years	CR ^8^/CRi ^10^: 80%; 75% MRD ^9^ negative	1-year estimated OS ^11^: 75%	30-day mortality: 0% 60-day mortality: 0%

^1^ s-AML = secondary acute myeloid leukemia. ^2^ t-AML = therapy-related acute myeloid leukemia. ^3^ AML = acute myeloid leukemia. ^4^ ts-AML = treated secondary acute myeloid leukemia. ^5^ MDS = myelodysplastic syndrome. ^6^ CRc = composite complete remission. ^7^ ORR = overall response rate. ^8^ CR = complete remission. ^9^ MRD = minimal residual disease. ^10^ CRi = complete remission with incomplete bone marrow recovery. ^11^ OS = overall survival. ^12^ EFS = event-free survival. ^13^ DFS= disease-free survival.

## 3. Venetoclax Plus Intensive Chemotherapy in Refractory/Resistant AML

The FLAG-IDA plus VEN regimen was investigated in an R/R-AML setting and showed encouraging preliminary results. After the starting 39 patients were treated [23], a protocol amendment recommended reducing the VEN-treatment days from 21 to 14 days and the cytarabine dose from 2 to 1.5 g/m^2^ (phase 2 cohort) due to the occurrence of severe grade 3 and grade 4 neutropenia-related infections in the initial phase Ib study. Therefore, further trials combining VEN plus IC regimens were designed with a shorter 7-day VEN regimen [29]. Among the R/R AML patients included in the FLAG-IDA plus VEN trial phases Ib (*n* = 16; median age, 51 years) and IIB (*n* = 23; median age, 47 years) [23], the CR + Cri rate was 67% (69% were MRD-negative), and 46% proceeded to consolidative HSCT. The estimated 1-year EFS and OS rates were 41% and 68%, respectively, which represent a significant outcome amelioration in comparison with historical results for R/R AML [33,34]. The rate of adverse events resembled that observed in de novo AML trials. Febrile neutropenia and bacteremia were observed in 51% and 46% of the patients, respectively, with an increased rate of sepsis documented in the phase 1b cohort compared with the phase 2 cohorts (50% versus 43%). The 30-day and 60-day mortality among this high-risk group was 0% and 4.4%, respectively.

A real-life experience retrospectively evaluated the outcomes of 24 R/R-AML patients (44% of whom had previously received HSCT) treated with the FLAG-IDA regimen plus VEN [35]. This resulted in a CR/CRi of 72% (91% in the post-HSCT group) with a 1-year DFS and OS of 67% and 50%, respectively. However, this approach appeared too toxic, showing a 30-day and 60-day mortality of 12% and 48%, respectively.

The FLA-IDA plus VEN (FLAVIDA) regimen (VEN given at 100 mg daily for 7 days when posaconazole was co-administered) was investigated in 13 R/R-AML patients who were predominantly in first-salvage therapy (number of previous lines of treatment: 1–5) and who were classified as ELN intermediate- and high-risk (84.6%) AML [36]. The ORR after induction was 69%, with a median time of CR/Cri maintenance of 7.3 months and an estimated 6-month EFS and OS of 52% and 76%, respectively. Afterward, a retrospective analysis of the same study analyzed the difference between 37 patients receiving FLAVIDA and a cohort of 81 patients treated with FLA-IDA but without VEN. The two cohorts were normalized according to age, genetic characteristics, R/R status, and previous HSCT [37]. Patients treated with the FLAVIDA regimen showed a more significantly improved ORR (78% vs. 47%, *p* = 0.001); however, no significant difference was observed in terms of the OS and bone-marrow recovery time post-therapy between the two groups.

An approach based on the combination of VEN with high-dose cytarabine and mitoxantrone (HAM) was investigated in the phase I/II Alliance Leukemia (SAL) Relax trial [38]. Twelve patients (median age: 57 years) presenting with relapsed AML were included in this dose-finding phase study. The combination of VEN 400 mg daily (after a 3-day ramp-up) on days 3 to 14 and HAM proved secure and effective, leading 11 out of 12 patients to achieve remission.

Finally, the administration of VEN in an R/R setting has also been explored in combination with CPX-351. A phase 2 trial enrolled 26 R/R-AML patients; among them, 10 (42%) had a complex karyotype and 6 (23%) were TP53-mutated AML [31]. The CR/Cri rate was 46%. Furthermore, 78% of CR/Cri patients achieved MRD negativity. Out of the 12 responding patients, 10 (83%) underwent HSCT. The 4- and 8-week mortality was 12% and 19%, respectively, with the most common grade 3 and grade 4 adverse events including infections, pneumonia, and febrile neutropenia. The 1-year estimated OS was 39%, while in responding patients, the median OS was 26.9 months. Table 2 summarizes these trials, including the combination of VEN and chemotherapy in an R/R setting.

## 4. Venetoclax Plus Intensive Chemotherapy in Pediatric AML

The first study combining VEN plus IC in a pediatric setting was conducted at the St. Jude Children’s Research Hospital. The trial enrolled 2- to-22-year-old R/R-AML patients, evaluating as the primary endpoint the safety and activity of VEN (240 or 360 mg/m^2^/day orally for 28 days) in combination with cytarabine (100 mg/m^2^ every 12 h for 20 doses or 1000 mg every 12 h for 8 doses) and with or without idarubicin (12 mg/m^2^) as a single dose. Results of this phase 1 study also indicated the recommended phase 2 dose of VEN plus IC [39]. In the following phase 2 trial, 38 patients were enrolled, and the recommended doses were 360 mg/m^2^ for VEN and 1000 mg/m^2^ (8 doses) for cytarabine. Overall responses were documented in 24 (69%) of 35 evaluable cases, comprising 16 CRs (11 MRD-negative). A large number (80%) of patients who reached CR/CRi, all of whom were MRD-negative after cycle 1, proceeded to a subsequent HSCT. The most frequent grade 3–4 toxicities resulted of febrile neutropenia (25 patients), bloodstream infections (6 patients), and invasive fungal infections (6 patients), with treatment-related death observed in one patient.

A recent retrospective analysis detailed the outcomes of pediatric patients with R/R AML who underwent VEN treatment before HSCT at St. Jude Children’s Research Hospital [40]. Twenty-five pediatric patients (median age: 13.1 years) received VEN-based salvage therapy (VEN/cytarabine, 15; VEN/cytarabine/idarubicin, 5; VEN/cytarabine/azacytidine, 3; or VEN/decitabine, 2) prior to HSCT. Notably, 9 of the 25 patients had already undergone a prior HSCT. The last dose of VEN was administered at a median of 19 days prior to the start of the HSCT conditioning regimen. At the time of the HSCT, 14 patients were in MRD-negative CR, 9 patients were MRD+ (≥0.01%), and 2 patients had active disease (bone marrow blasts > 5%). At a median follow-up of 280 days from HSCT, the 1-year overall survival (OS) was 80% and leukemia-free survival was 74.5%. Unfortunately, six patients relapsed at a median of 143 days, and only one patient experienced non-relapse mortality.

An Italian multicenter retrospective analysis of pediatric patients (0–18 years) with R/R AML or advanced MDS arising after chemotherapy or radiation therapy who received VEN-based combination therapies was reported. Thirty-one patients (median age: 10.2 years) experiencing a median of three previous lines of therapy (31.2% relapsed after HSCT) were included in the analysis [41]. The rates of CR, partial response, and no response were 66.7%, 11.1%, and 22.2% of cases, respectively. Of note, patients who received VEN with hypomethylating agents achieved CR, a partial response, no response, and treatment failure in 36.8%, 26.3%, 31.6%, and 5.3% of cases, respectively. Twenty patients (64%) were successfully bridged to HSCT by VEN therapy after a median time of 3.3 months from the start of the VEN treatment. The estimated 30-month OS after the start of VEN was 29.9% for the whole cohort and 74.4% for patients undergoing HSCT.

According to this data, the combination of VEN plus IC therapy demonstrates efficacy in the pediatric setting for AML, with a tolerability profile similar to that observed in adult AML. Nevertheless, additional data are needed to substantiate the actual benefits of this approach and to assess its potential role in frontline regimens.

## 5. Molecular Markers Predicting the Response to Venetoclax and Chemotherapy-Based Regimens

The combination of VEN plus IC demonstrates significant efficacy across various genetic subgroups in AML. In patients with ELN favorable-, intermediate-, or unfavorable-risk AML, frontline treatment with FLAG-IDA plus VEN yields composite CR rates of 88%, 89%, and 89%, respectively [24]. Similarly, patients receiving CLIA plus VEN for ELN favorable-, moderate-, or high-risk AML exhibit 1-year survival rates of 78%, 93%, and 81%, respectively [29]. The efficacy of the regimen including VEN plus “2 + 6” daunorubicin and cytarabine is evident across the ELN2022 risk groups. For the favorable-risk group, the estimated 12-month OS, EFS, and DFS rates are 87.5%, 88.9%, and 100%, respectively. The intermediate-risk patients show a rate of 100% for all three endpoints, while adverse-risk patients exhibit a rate of 70.7%, 70.3%, and 79.9%, respectively [18]. The FLAG-IDA plus VEN regimen also showed efficacy in patients with extramedullary localization of the disease; indeed, among patients with extramedullary AML (three with de novo AML and one with R/R AML), all had durable responses, with three going on to HSCT [24].

According to the CAVEAT trial, a seven-day VEN pre-phase before combining the “5 + 2” regimen resulted in a significant reduction in bone marrow blasts in patients with de novo AML, especially in those patients with *NPM1* (56% reduction), *IDH2* (55% reduction), or *SRSF2* (47% reduction) mutations. As a result, patients with these mutations showed an encouraging median OS (*NPM1*: 13.2 months; *IDH2*: not reached; and *SRSF2*: 31.3 months) [19]. In a subsequent analysis of the CAVEAT study evaluating the TFR duration in patients showing CR/Cri, 75% of patients with an *NPM1* or *IDH2* mutation at diagnosis were in TFR [19]. Notably, R/R patients presenting with *NPM1, IDH1*, and *IDH2* mutations also showed promising responses when they received an intensive combination treatment with VEN. Indeed, the FLAG-IDA plus VEN approach produced 100% composite CR in these molecular subgroups, thereby allowing 71% of the patients to undergo HSCT [23]. Also, *KMT2A*-rearranged patients showed a 100% composite CR rate (80% MRD-negativity by PCR for *KMT2A*) after FLAG-IDA plus VEN, with a consequent 1-year OS of 80% [23]. However, in a pediatric trial [39], among 12 patients with *KMT2A* rearrangements, 5 patients showed no response.

While FLAG-IDA plus VEN did not predict the outcome of patients presenting with signaling pathway gene mutations (*K/NRAS*, *PTPN11*, *FLT3*, *CBL*, and *KIT*) in the frontline setting, R/R-AML patients with these mutations showed a lower median OS than patients with wild-type signaling genes. Similar findings were noted in R/R-AML patients, where tumor-suppressor gene mutations (*TP53*, *WT1*, *FBXW7*, or *PHF6*) were linked to a considerably lower rate of composite CR (38%) and a worse prognosis (median OS: 7 months) [23].

In the CAVEAT trial, the lowest blast reductions after VEN pre-phase treatment were observed in patients with *TP53* and kinase-activating mutations (*FLT3-ITD*, *FLT3-TKD*, *RAS*, and *PTPN11*). Consequently, dismal OS outcomes were observed in patients with *TP53* (3.6 months) and *FLT3-ITD* (5.5 months) mutations [19]. In a CLIA plus VEN trial [29], 15 patients had a *FLT3-ITD* and/or -*TKD* mutation, and 9 of these patients received a concomitant FLT3 inhibitor during induction and consolidation. Patients receiving a concomitant FLT3 inhibitor had a similar OS (*p* = 0.38) and EFS (*p* = 0.20) compared with those receiving VEN plus CLIA alone. In the FLAG-IDA plus VEN trial, 10 patients (3 with de novo AML and 7 with R/R AML) showed *TP53* mutations at baseline. The overall composite CR rate was 60%, with a median duration of response and OS of 3.4 and 9 months in de novo AML and 3.2 and 7 months in R/R AML, respectively [23].

Recently, data from VIALE-A demonstrated that AML patients without specific genetic mutations (*FLT3-ITD*, *K/RAS*, *N/RAS*, and *TP53* mutations) may derive a greater benefit from the combination of VEN and AZA [42]. In this subgroup of patients, the median OS was reported to be 26 months, which represents a significant improvement compared with traditional treatment approaches. The identification of specific molecular profiles that predict a higher benefit from VEN plus IC treatments is an important goal in AML research. Indeed, the possibility of tailoring treatment based on a patient’s genetic and molecular characteristics can help to optimize outcomes and to minimize potential side effects.

## 6. Conclusions

Several lines of evidence support the clinical efficacy of VEN-based approaches in both induction and consolidation therapies for AML. In a retrospective analysis that specifically focused on younger patients newly diagnosed with AML (with median age of 49 years), a propensity-score-matching analysis revealed that the addition of VEN plus IC resulted in higher rates of MRD-negative remission and facilitated the safe transition to consolidative HSCT for a significant number of patients [30]. Furthermore, the combination of VEN plus IC not only improved the EFS for the entire study population but also demonstrated a notable enhancement in OS, particularly in patients categorized as having an intermediate or adverse AML risk according to the ELN classification. Significantly, the magnitude of improvement in EFS and OS was more pronounced in patients classified as adverse-risk AML. However, differences in patient selection and clinical profiles across different trials might have yielded conflicting results.

In a recent meta-analysis conducted by Shimony et al. [43], nine studies involving VEN plus IC for de novo AML were included. The pooled ORR across all studies was 86.2%, with a combined CR rate of 69%. The MRD rate among responders (reported in only seven studies) showed a combined MRD-negativity rate of 79.4%.

In contrast, frontline VEN-based treatment regimens do not seem to improve the clinical outcome of TP53-mutated AML [19,23]. Apart from patients who have *TP53* mutations, an improved outcome was observed in other adverse-risk molecular subgroups. This includes patients with signaling mutations (such as *RAS*, *FLT3-ITD* or *FLT3-TKD*, and *PTPN11*) or splicing mutations (like *SRSF2*, *U2AF1*, *SF3B1*, and *ZRSR2*) [23,24]. Patients with splicing mutations demonstrated outcomes similar to those observed in s-AML. Since s-AML patients show improved OS when treated with VEN plus IC compared with IC alone, it is reasonable to consider the former regimen as a possible standard-of-care. Results from an ongoing phase 3 clinical trial (NCT04628026) assessing the effectiveness of a treatment regimen involving “7 + 3” chemotherapy versus “7 + 3” chemotherapy combined with VEN are eagerly waited.

In the context of R/R AML, the elevated rate of MRD-negative responses achieved, together with the successful transition to HSCT in patients receiving FLAG-IDA plus VEN, provides robust efficacy data for this regimen [23]. Therefore, this treatment should be contemplated alongside other traditional salvage regimens like FLAG-IDA and MEC.

Another important issue to be further addressed is whether HSCT might be unnecessary in specific categories of cases who achieve MRD-negative responses, especially those with ELN intermediate-risk AML. Indeed, the clear-cut benefit of consolidative HSCT in this context is yet to be defined. Recent data indicated that there was no significant difference in EFS or OS when HSCT was adopted in patients who had previously received FLAG-IDA combined with VEN [24]. Whether these intensive VEN-based induction approaches can determine deeper responses and be potentially curative for selected patients, thus sparing consolidation with HSCT, is a is question whose answer highly anticipated.

Results of ongoing clinical investigations are expected to provide valuable insights into the ideal dose and duration when combining VEN with IC (Table 3). Moreover, in the absence of guidelines, collaboration among academic leukemia centers is recommended for the optimal design of studies analyzing the role of IC-venetoclax combinations in younger AML patients. This research should aim to minimize adverse effects, such as myelosuppression and infectious events, while ensuring treatment efficacy. In parallel, translational studies should help to understand the complex cellular and molecular events contributing to resistance to VEN plus IC regimens. This knowledge will hopefully allow therapies to be tailored on an individual basis. The expanded results from these current early-phase trials, as well as those trials incorporating additional targeted therapies alongside VEN, will play a crucial role in shaping the future of AML treatment.

## Figures and Tables

**Figure 1 cancers-16-00073-f001:**
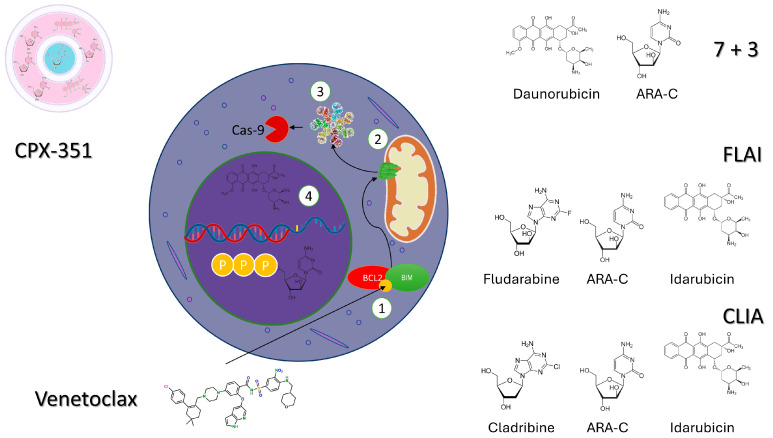
Vignette showing the potential mechanisms of action of intensive CHT combined with venetoclax. (1) Venetoclax is a member of a class of BH3-mimetic drugs. The BH3 domain of the anti-apoptotic protein is shown in yellow. When venetoclax is administered to AML patients, it can displace the activator protein BIM (or BID). Then, the activator protein of apoptosis in combination with the effector (BAX, BAK) proteins results in the release of cytochrome C from the outer mitochondrial membrane (2). (3) The efflux of cytochrome C promotes the formation of an apoptosome, which, in turn, activates Caspase 9 and a caspase cascade that culminates in apoptosis. The simultaneous administration of intensive chemotherapy generates intense stress and damage to nuclear DNA. In particular, (4) daunorubicin damages DNA by intercalating between base pairs, resulting in an uncoiling of the helix, ultimately inhibiting DNA synthesis and DNA-dependent RNA synthesis. At the same time, ARA-C is phosphorylated three times in the cytoplasm and enters the nucleus, where it is incorporated into DNA, arresting DNA polymerase.

**Table 2 cancers-16-00073-t002:** Trials including a venetoclax and chemotherapy combination in relapsed/refractory AML.

Trial/Reference	Design	Primary Endpoint	Number of Patients	Patients Enrolled	Response	Outcomes	Early Mortality
fludarabine + cytarabine + idarubicin + filgastrim + venetoclax (FLAG-IDA+ VEN)/[23]	phase Ib/II	overall response rate	49	patients aged ≥ 18	CR ^1^/CRi ^2^ rate: 67% (69% were MRD ^3^-negative); 46% proceeded to HSCT ^4^	estimated 1-year EFS ^7^: 41% estimated 1-year OS ^8^: 68%	30-day mortality: 0% 60-day mortality: 4.4%
fludarabine + cytarabine + idarubicin + filgastrim + venetoclax (FLAG-IDA + VEN)/[35]	real-life analysis	/	24	patients aged ≥ 18	CRc ^5^: 72% for the entire cohort and 91% in patients treated for post-HCT ^4^ relapse	12-month RFS ^9^: 67% 12-month OS ^8^: 50%	30-day mortality: 12%; 60-day mortality: 48%
fludarabine + cytarabine + idarubicin + venetoclax (FLAVIDA)/[36]	real-life analysis	/	13	patients aged ≥ 18	ORR ^6^: 69%, with a median duration of CR ^1^/CRi ^2^ of 7.3 months	estimated 6-month EFS ^7^: 52% estimated 6-month OS ^8^: 76%,	/
high-dose cytarabine + mitoxantrone + venetoclax (HAM + VEN)/[38]	phase I/II	dose-limiting toxicity	12	patients aged ≥ 18	CR ^1^/CRi ^2^ rate: 92% (62.5% were MRD ^3^-negative);	/	30-day mortality: 8.3%
CPX-351 + venetoclax (CPX-351 + VEN)/[31]	phase Ib/II	the safe dose and schedule	26	patients aged ≥ 18	CR ^1^/CRi ^2^ rate: 46% (78% were MRD ^3^-negative).	1-year estimated OS ^8^: 39%; in responding patients, the median OS ^8^ was 26.9 months	30-day mortality: 12% 60-day mortality: 19%
+ cytarabine +/− idarubicin + venetoclax/[39]	phase Ib/II	the safe dose and schedule	38	patients between 2 and 22 years	ORR ^6^: 69%; 80% of patients who achieved a CR ^1^/CRi ^2^ proceeded to HSCT ^4^	/	/

^1^ CR = complete remission. ^2^ Cri = complete remission with incomplete bone marrow recovery. ^3^ MRD = minimal residual disease. ^4^ HSCT = hematopoietic stem cell transplantation. ^5^ CRc = composite complete remission. ^6^ ORR = overall response rate. ^7^ EFS = event-free survival. ^8^ OS = overall survival. ^9^ RFS = relapse-free survival.

**Table 3 cancers-16-00073-t003:** Ongoing recruiting trials comprising intensive chemotherapy (IC) and venetoclax combinations.

Clinical Trial Gov. Identifier	Trial Phase	Combination Treatment	Treatment Status
NCT06068621	2	CACAG ^1^ + venetoclax	Newly diagnosed AML ^2^
NCT05356169	2	DA ^3^ “3 + 7” + venetoclax	Newly diagnosed AML ^2^
NCT05522192	1/2	Mitoxantrone hydrochloride Liposome + venetoclax	Relapsed/Refractory AML ^2^
NCT06084819	2	CACAG ^1^ + venetoclax	Relapsed/Refractory AML ^2^
NCT05918198	2	CAG ^4^ + venetoclax	Relapsed/Refractory AML ^2^
NCT05807347	2	Azacytidine + CAG ^4^ + venetoclax	Relapsed/Refractory AML ^2^
NCT03629171	2	CPX351 + venetoclax	Newly diagnosed AML ^2^ and relapsed/refractory AML ^2^
NCT05780879	2	FLAG ^5^ or CLAG ^6^ + venetoclax	Newly diagnosed AML ^2^
NCT05263284	1	8-Chloroadenosine + venetoclax	Relapsed/Refractory AML ^2^
NCT03709758	2	DA ^3^ + venetoclax	Newly diagnosed AML ^2^
NCT05805098	2/3	Homoharringtonine + cytarabine + venetoclax	Newly diagnosed AML ^2^
NCT03826992	1	CPX351 + venetoclax	Relapsed/Refractory AML ^2^
NCT02115295	2	CLIA + venetoclax	Newly diagnosed AML ^2^ and relapsed/refractory AML ^2^
NCT04797767	1	CLAG-M ^7^ + venetoclax	Newly diagnosed AML ^2^ and relapsed/refractory AML ^2^

^1^ CACAG: azacytidine + cytarabine+ aclacinomycin + chidamide + granulocyte colony-stimulating factor. ^2^ AML: acute myeloid leukemia. ^3^ DA: daunorubicin + cytarabine. ^4^ CAG: cytarabine + aclacinomycin + granulocyte colony-stimulating factor. ^5^ FLAG: fludarabine + cytarabine + granulocyte colony-stimulating factor. ^6^ CLAG: cladribine + cytarabine + granulocyte colony-stimulating factor. ^7^ CLAG-M: G-CSF, cladribine, cytarabine, and mitoxantrone.

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
