# Peer review of "Venetoclax: A Game Changer in the Treatment of Younger AML Patients?"

_cancers, 2023, doi:10.3390/cancers16010073_

Round 1
Reviewer 1 Report
Comments and Suggestions for Authors
In this review paper Molica et al present ongoing trials that involve the addition of VEN to intensive chemotherapy in frontline protocols , with a focus on evaluating its efficacy in younger patients with AML, and specific molecularly defined subgroups. In general, the paper is well written, scientifically accurate, and clinically valid.
1.Page 2, line 54 – measurable residual disease should be used
2.The subchapter and additional with ongoing clinical trials with venetoclax combinations in younger AML patients should be added
Author Response
1) measurable residual disease was introduced in the text
2) we added a new table (table 3) including the ongoing trials which provide the combination of intensive chemotherapy and venetoclax in de novo and R/R AML
Reviewer 2 Report
Comments and Suggestions for Authors
The document is a review of clinical trials on venetoclax combined with chemotherapy in younger Acute Myeloid Leukemia (AML) patients. It highlights venetoclax's effectiveness in improving response rates and aiding the transition to curative treatments, particularly for high-risk AML patients, and underscores its role in enhancing chemotherapy across various risk groups, offering insights for future research.
A few suggestions to improve the manuscript:
Clarify the selection criteria for clinical trials included in the review. More detailed explanations of inclusion and exclusion criteria will enhance the transparency of the review process.
Address any conflicting results from different studies. This could involve a more in-depth analysis of why certain studies might have yielded different outcomes.
In conclusion remarks, the authors could provide more recommendations for integrating venetoclax into clinical practice, including potential guidelines or criteria for selecting suitable patients.
Offer guidance on what the findings mean for future research directions, including potential studies that could build on these results.
Author Response
1) At the end of the indroduction, we clarified the selection criteria for clinical trials in the review
2) We added a sentence in conclusions paragrph
3 and 4) We added a sentence in conclusions paragraph
Reviewer 3 Report
Comments and Suggestions for Authors
In this review, authors discuss current studies assessing the efficacy of frontline regimens integrating Venetoclax into intensive chemotherapy regimens in younger patients with AML. The review is clear and well written. Some few suggestions are listed below:
- Please verify the number of patients on Table 1, first line :36 in the table, 33 on line 91
- Please add more details on venetoclax administered dose: line 161
- Please specify the age (median or range) of patients enrolled in the GIMEMA AML1718 trial in both table and results.
- Please add 60-days mortality in flu+cyta+ida +filgrastim+ ven in table 2 as detailed in the text: line 267
- Please specify Idarubicine dose/day of administration in the pediatric trial (line 305)
Author Response
we added all the suggestions